# Species identification and phylogenetic analysis of *Leishmania* isolated from patients, vectors and hares in the Xinjiang Autonomous Region, The People's Republic of China

**Yun-Fu Chen**[1☯], **Li-Fu Liao**[2☯], **Na Wu**[1], **Jiang-Mei Gao**[1,3], **Peng Zhang**[1], **Yan-Zi Wen**[1], **Geoff Hide**[4], **De-Hua Lai**[1]*, **Zhao-Rong Lun**[1,4]*

**1** MOE Key Laboratory of Gene Function and Regulation, State Key Laboratory of Biocontrol, School of Life Sciences, Sun Yat-Sen University, Guangzhou, The People's Republic of China, **2** Center for Laboratory Animal Research, Xinjiang Uighur Autonomous Region Center for Disease Control and Prevention, Urumqi, The People's Republic of China, **3** Guangdong Key Laboratory of Animal Conservation and Resource Utilization, Institute of zoology, Guangdong Academy of Sciences, Guangzhou, The People's Republic of China, **4** Ecosystems and Environment Research Centre and Biomedical Research Centre, School of Science, Engineering and Environment, University of Salford, Salford, United Kingdom

☯ These authors contributed equally to this work.
* laidehua@mail.sysu.edu.cn (D-HL); lsslzr@mail.sysu.edu.cn (Z-RL)

**Data Availability Statement:** All relevant data are within the manuscript.

## Abstract

### Background

Visceral leishmaniasis (VL) has been declared as one of the six major tropical diseases by the World Health Organization. This disease has been successfully controlled in China, except for some areas in the western region, such as the Xinjiang Autonomous Region, where both anthroponotic VL (AVL) and desert type zoonotic VL (DT-ZVL) remain endemic with sporadic epidemics.

### Methodology/Principal findings

Here, an eleven-year survey (2004–2014) of *Leishmania* species, encompassing both VL types isolated from patients, sand-fly vectors and Tarim hares (*Lepus yarkandensis*) from the Xinjiang Autonomous Region was conducted, with a special emphasis on the hares as a potential reservoir animal for DT-ZVL. Key diagnostic genes, ITS1, *hsp70* and *nagt* (encoding *N*-acetylglucosamine-1-phosphate transferase) were used for phylogenetic analyses, placing all Xinjiang isolates into one clade of the *L. donovani* complex. Unexpectedly, AVL isolates were found to be closely related to *L. infantum*, while DT-ZVL isolates were closer to *L. donovani*. Unrooted parsimony networks of haplotypes for these isolates also revealed their relationship.

### Conclusions/Significance

The above analyses of the DT-ZVL isolates suggested their geographic isolation and independent evolution. The sequence identity of isolates from patients, vectors and the Tarim

**Funding:** Laboratory is receiving the financial support from the National Natural Science Foundation of China (31720103918 and 31672276) to Z-RL (http://www.nsfc.gov.cn) and the Natural Science Foundation of Guangdong Province (2018A030313187) to YZW (http://pro.gdstc.gd.gov.cn). The funders had no role in study design, data collection and analysis, decision to publish, or preparation of the manuscript.

**Competing interests:** The authors have declared that no competing interests exist.

hares in a single DT-ZVL site provides strong evidence in support of this species as an animal reservoir.

## Author summary

Black faver, also known as visceral leishmaniasis (VL), is caused by pathogens of *Leishmania* species, spread by the bites of infected sand flies. This disease has been successfully controlled in China, except for some areas in the western region, such as Xinjiang. However, the knowledge on *Leishmania* in these areas remains a few important gaps. Particularly, what is the animal reservoir for desert type zoonotic VL (DT-ZVL), as sand flies get infected in areas free of patients or infected dogs? To address this question, an eleven-year survey (2004–2014) in Xinjiang for *Leishmania* species was carried out. We found that VLs in Xinjiang are contributed to *Leishmania donovani* complex, and Tarim hares is likely the reservoir animal for DT-ZVL.

## Introduction

Visceral leishmaniasis (VL), caused by species of the *Leishmania donovani* complex, is a potentially fatal disease if not treated [1]. In China, before the implementation of the national infectious diseases control programs in 1951, it was one of the major parasitic diseases. A total of approximately half a million VL cases were then reported in 16 provinces north of the Yangtze River [2]. The control programs have successfully eliminated VL in the northeastern plain, but not in the west and northwest regions, where this disease has persisted as three different types: mountain type zoonotic VL (MT-ZVL), anthroponotic VL (AVL) and desert type zoonotic VL (DT-ZVL) [2]. The total number of cases reported from these areas between 2002 and 2011 were 3,169 VL cases, ranging from 140 to 509 cases per year. This study considers the causative agents of AVL and DT-ZVL from the Xinjiang Autonomous Region in the northwest of China.

One old endemic site in Xinjiang is the Kashgar alluvial plain and the Aksu oasis, where the AVL is an endemic disease, whose pathogen is transmitted by the bites of the peridomestic vector *Phlebotomus longiductus*. Most patients are over 6 years old (70%) [3]. This type of VL has been considered as anthroponotic on account of its familial clustering and the uncertainty about the existence of potential animal reservoirs [4]. As such, the causative agent has long been referred to as *L. donovani*. More recently, isolates collected from this endemic region, have been identified as *L. infantum* based on the *nagt* and other single-copy gene analyses (isolates HOM/CN/91/911, HOM/CN/92/921, HOM/CN/97/9701) [5] and the multilocus sequence typing (MLST) of five enzyme-coding genes (e.g. isolate MHOM/CN/80/801) [6].

DT-ZVL is endemic in the northwestern China, mainly in the Bachu and Jiashi counties of Xinjiang, but also in the western part of Inner Mongolia and northern Gansu [3], with wild sand fly species *Phlebotomus wui* and *P. alexandri* as vectors specific for the sandy desert and pebble desert subtypes, respectively. Most patients are infants of 2 years-old or younger (94%). DT-ZVL is considered zoonotic and its pathogen is transmitted by wild sand fly species, *Phlebotomus wui* and *P. alexandri*, which live in rodent burrows as their natural habitats. This is further supported by the lack of familial clustering. Extensive studies in desert rodents, e. g. the great gerbil, have failed to establish their role as a reservoir animal for DT-ZVL, but have implicated the Tarim hare (*Lepus yarkandensis*) as a potential reservoir for DT-ZVL [7]. These

wild hares are unique to the Tarim basin in the Taklamakan desert. They were found to be seropositive for *Leishmania* antigens (rk39) at a high prevalence (25%), microscopically positive for amastigotes in infected tissues and to suffer from the typical VL symptom of spleno-megaly, hepatomegaly and sometimes ulcerative lesions of the ears. Moreover, promastigotes were isolated from the spleens of infected hares and found to be virulent for the laboratory-reared susceptible steppe rodent, *Lagurus lagurus*. The DNA (*nagt* locus) sequence identity of hare-, vector- and patient-derived isolates provided the preliminary evidence that the hare could be a potential reservoir of DT-ZVL [7]. The causative agent was considered to be *L. infantum*, since all the *nagt* sequences were identical to those of the reference *L. infantum* isolates from other endemic areas in China and elsewhere [5]. This was further confirmed by the ITS1 locus analysis [8]. In contrast, investigation of similar isolates with the same designated code names (MHOM/CN/00/Wangjie1 as a reference strain) classified them as *L. donovani* based on the ITS1 sequence [9] and MLST analyses [6].

The heterogeneity of Chinese *Leishmania* isolates was first observed by isoenzyme analysis and by DNA hybridization (kinetoplast and nuclear DNA) [10–12]. More recently, the diversity of the Chinese *Leishmania* isolates has been addressed by several other studies using various molecular markers [6,9,13–16]. By analysis of the molecular fingerprints, it was found that the isolates from the three epidemiological disease types were distinguishable (reviewed by [17]). However, there were limitations in these studies. Firstly, only a limited number of isolates were available for each given disease type. Secondly, the parasite samples did not represent the complete transmission cycle. Thirdly, only one isolate (IPHL/CN/77/XJ771) from the sand fly vector found in the DT-ZVL region has been studied thus far. Therefore, the species and their relationship with the epidemiological cycles in the regions of Xinjiang remained unclear. This was especially the case for DT-ZVL which until now had an unknown animal reservoir.

To resolve this dilemma, 20 *Leishmania* isolates were collected from endemic foci of both AVL and DT-ZVL in Xinjiang and analyzed using three different phylogenetic markers (*nagt*, ITS1 and *hsp70*). Our analyses of representative isolates cultured from captured hares, patients and the vector *P. wui* strongly supports the conclusion that the Tarim hare serves as a unique reservoir of DT-ZVL.

## Materials and methods

### Ethics statement

Samples collected from patients were approved by The Ethical Committee of Xinjiang Uighur Autonomous Region Center for Disease Control and Prevention under license of #30460120. Formal verbal consent was obtained from each patient or parent of child patient.

### Parasite collection and cultivation

Samples investigated in this study were collected in Bachu, Jiashi, Minfeng and Shufu counties of Xinjiang province, northwestern China, during an 11 year survey of the VL in the endemic regions of the Tarim Basin (Fig 1 and Table 1). In order to isolate and maintain parasites, bone marrow aspirates from patients, dissected gut of infected sand-flies and homogenates of the lesion or spleen tissue from infected Tarim hares were inoculated into the *Lagurus lagurus* or the grey hamsters. Parasites were recovered from spleen homogenates of infected animals and cultured in modified medium LLM at 27˚C [7]. Parasites or spleen samples were stored at 70% ethanol before DNA extraction. Protocols for the use of animals were approved by The Center for Laboratory Animal Research of Xinjiang and Institutional Review Board for Animal Care at Sun Yat-Sen University.

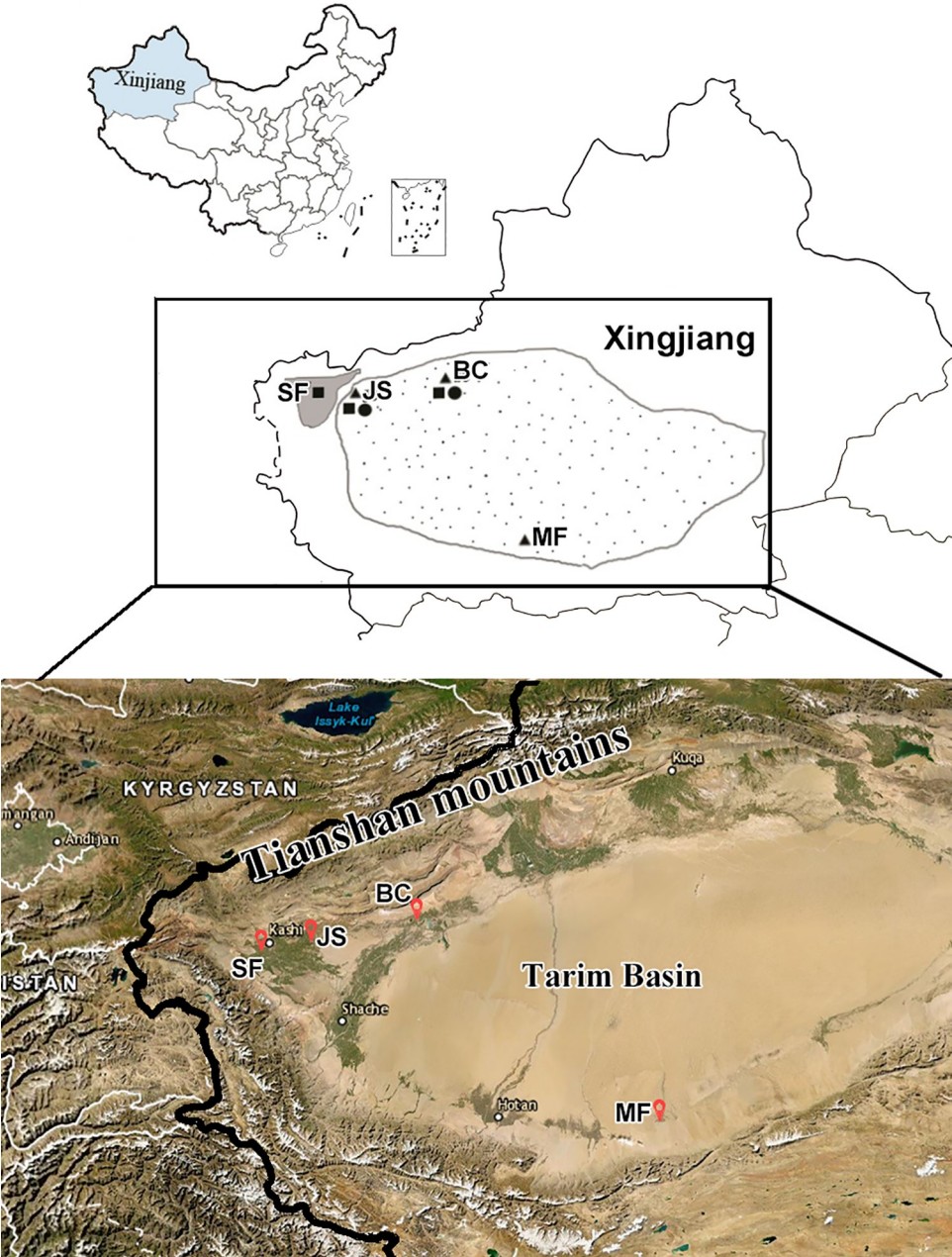

**Fig 1. The sampling areas in the Xinjiang Autonomous Region.** *Leishmania* samples were collected from oases (grey area) or desert (spotted area) (Upper Panel). SF, Shufu county; JS, eastern Jiashi county; BC, Bachu county; MF, Minfeng county. Black square: Patient, Black circle: Tarim hare, Black triangles: *P.wui* (vector). The Tarim basin is surrounded by the Tianshan mountains and Kunlun mountains which forms a barrier separating it from surrounding areas. https://usgs.maps.arcgis.com/apps/webappviewer/index.html?id=2b7d5dbc0be340b9b17f6d94aac5b713 (42.8–35.0˚N, 71.4–87.6˚E).

## DNA extraction, PCR amplification and sequencing of the ITS1, *hsp70* and *nagt* region

Sample DNA was extracted as described elsewhere [18]. The ITS1, *hsp70* and *nagt* fragments were amplified by using the following primer sets, ITS1: L5.8S (5' TGA TAC CAC TTA TCG

**Table 1.** *Leishmania* isolates from Xinjiang.

| Name | WHO code | Source/Host (Age) | Time of isolation | Location* | Type | Haplotype | | Genbank accession No. (ITS1, *hsp70, nagt*) |
|---|---|---|---|---|---|---|---|---|
| | | | | | | ITS1 | *hsp70* | |
| 934 | MHOM/CN/2004/KBC-1 | Patient (11 months) | Dec. 2004 | Bachu | DT-ZVL | I2 | H2 | KU975140.1, KX150477.1, KX150463.1 |
| 2703 | IMJW/CN/2005/KBC-2 | *P.wui* | Jun. 2005 | Bachu | DT-ZVL | I2 | - | KU975141.1, NA, NA |
| 2681 | MLEP/CN/2005/KBC-1 | Tarim hare | Dec. 2005 | Bachu | DT-ZVL | I2 | - | KU975142.1, NA, NA |
| 418 | MLEP/CN/2004/KBC-2 | Tarim hare | Dec. 2004 | Bachu | DT-ZVL | I2 | H2 | KU975143.1, KX150478.1, KX150464.1 |
| 432 | MLEP/CN/2004/KBC-3 | Tarim hare | Dec. 2004 | Bachu | DT-ZVL | I2 | H2 | KU975144.1, KX150479.1, NA |
| 3153 | MHOM/CN/2009/KJS-1 | Patient (1 year) | Jan. 2009 | Jashi | DT-ZVL | I2 | H2 | KU975145.1, KX150480.1, KX150465.1 |
| 3081 | MHOM/CN/2009/KJS-3 | Patient (11 months) | Jan. 2009 | Jashi | DT-ZVL | I2 | H2 | KU975146.1, KX150481.1, NA |
| 3208 | MHOM/CN/2009/KJS-4 | Patient (7 months) | Feb. 2009 | Jashi | DT-ZVL | I2 | - | KU975147.1, NA, NA |
| 3227 | MHOM/CN/2009/KJS-5 | Patient (7 months) | Nov. 2009 | Jashi | DT-ZVL | I2 | H2 | KU975148.1, KX150482.1, KX150466.1 |
| 3228 | MHOM/CN/2009/KJS-6 | Patient (6 months) | Nov. 2009 | Jashi | DT-ZVL | I2 | H2 | KU975149.1, KX150483.1, KX150467.1 |
| 2693 | IMJW/CN/2008/KJS-1 | *P.wui* | Jul. 2008 | Jashi | DT-ZVL | I2 | H2 | KU975150.1, KX150484.1, KX150468.1 |
| 3044 | MLEP/CN/2007/KJS-1 | Tarim hare | Dec. 2007 | Jashi | DT-ZVL | I2 | H2 | KU975151.1, KX150485.1, KX150469.1 |
| 3410 | IMJW/CN/2014/HMF-1 | *P.wui* | Sep. 2014 | Minfeng | DT-ZVL | I2 | H2 | KU975152.1, KX150486.1, KX150470.1 |
| 3416 | IMJW/CN/2014/HMF-2 | *P.wui* | Sep. 2014 | Minfeng | DT-ZVL | I2 | H2 | KU975153.1, KX150487.1, KX150471.1 |
| 3009 | MHOM/CN/2009/KSF-1 | Patient (2 years) | Jan. 2009 | Shufu | AVL | I1 | H1 | KU975154.1, KX150488.1,NA |
| 2616 | MHOM/CN/2009/KSF-2 | Patient (24 years) | Feb. 2009 | Shufu | AVL | I1 | H1 | KU975155.1, KX150489.1, KX150472.1 |
| 3149 | MHOM/CN/2009/KSF-3 | Patient (10 years) | Mar. 2009 | Shufu | AVL | I1 | H1 | KU975156.1, KX150490.1, KX150473.1 |
| 2618 | MHOM/CN/2009/KSF-4 | Patient (53 years) | Mar. 2009 | Shufu | AVL | I1 | H1 | KU975157.1, KX150491.1, KX150474.1 |
| 3219 | MHOM/CN/2009/KSF-6 | Patient (4 years) | May. 2009 | Shufu | AVL | I1 | H1 | KU975158.1, KX150492.1, KX150475.1 |
| 3344 | MHOM/CN/2009/KSF-7 | Patient (3 years) | Feb. 2009 | Shufu | AVL | I1 | - | KU975159.1, NA, NA |

* Bachu, Jashi, Minfeng and Shufu are counties in Xinjiang Autonomous Region, the People's Republic of China.

-, not available; NA, no amplicon.

CAC TT 3') and LITSR: (5' CTG GAT CAT TTT CCG ATG 3') [19]; *hsp70*: F25 (5' GGA CGC CGG CAC GAT TKC T 3') and R1310 (5' CCT GGT TGT TGT TCA GCC ACTC 3') [20]; *nagt*: L1(5' TCA TGA CTC TTG GCC TGG TAG 3') and L4 (5' CTC TAG CGC ACT TCA TCG TAG 3') using standard conditions as published [5]. PCR products were sequenced by Invitrogen (Life technology) and all sequences were deposited in GenBank under the accession numbers provided in **Table 1**.

## Sequence alignment and phylogenetic analysis

The sequences were checked manually and aligned with a set of *Leishmania* strains retrieved from GenBank (Table 2) using MEGA, version 5.0 [21] The unaligned 5' and 3' ends were removed before phylogenetic analysis. An imported Iranian *L. major* strain (MHOM/CN/2015/CPOLM-1) collected from a patient found in Guangzhou was used as an outgroup in the ITS1 analysis [22]. The phylogenetic relationships among the isolates were inferred from the phylogenetic tree reconstruction by the Neighbor Joining (NJ) using MEGA 5.0 by default setting and the reliability of the internal branches was tested by 1,000 bootstrap replications.

## Haplotype networks

Unrooted parsimony networks of haplotypes for *L. donovani* complex were constructed using TCS v.1.21 [23], with gaps treated as a fifth state.

**Table 2. List of reference strains used in this study.**

| Species | WHO code | Origin | Gene | Haplotype number | Accession |
|---|---|---|---|---|---|
| *L. donovani* | MHOM/CN/00/Wangjie1 | China | ITS1 | I2 | AJ000294 |
| *L. donovani* | IPHL/CN/77/XJ771 | Bachu, China | ITS1 | I2 | HM130608 |
| *L. donovani* | MCAN/CN/60/GS1 | Gansu, China | ITS1 | I5 | HQ830354 |
| *L. donovani* | MHOM/ET/67/HU3 | Ethiopia | ITS1 | I3 | AJ634373 |
| *L. donovani* | MHOM/IN/80/DD8 | India | ITS1 | I4 | AJ000292 |
| *L. donovani* | MHOM/SD/93/9S | Sudan | ITS1 | I3 | AJ634372 |
| *L. donovani* | MHOM/IQ/1981/SUKKAR2 | Iraq | ITS1 | I6 | AM901452 |
| *L. donovani* | MHOM/KE/83/NLB189 | Kenya | ITS1 | I4 | AJ634374 |
| *L. infantum** | MHOM/CN/08/Jiashi-1 | Jiashi, China | ITS1 | I2 | GQ367486 |
| *L. infantum* | MHOM/CN/54/Peking | Beijing, China | ITS1 | I1 | AJ634345 |
| *L. infantum* | MHOM/CN/78/D2 | Xinjiang, China | ITS1 | I1 | AJ000303 |
| *L. infantum* | MHOM/TN/80/IPT1 | Tunisia | ITS1 | I1 | AJ000289 |
| *L. infantum* | MHOM/FR/78/LEM75 | France | ITS1 | I1 | AJ634339 |
| *L. infantum* | MCAN/ES/86/LEM935 | Spain | ITS1 | I1 | AJ634355 |
| *L. infantum* | MHOM/IT/94/ISS1036 | Italy | ITS1 | I1 | AJ634353 |
| *L. infantum* | MHOM/IR/2012/Savodjbolagh11 | Iran | ITS1 | I1 | KC347299 |
| *L. infatum* | MHOM/UZ/2007/KU | Uzbekistan | ITS1 | I1 | FM164420 |
| *L. major* | MHOM/CN/2015/CPOLM-1 | China, imported | ITS1 | - | KU975160 |
| *L. donovani* | MHOM/CN/00/Wangjie1 | China | *hsp70* | H2 | HF586394 |
| *L. donovani* | MHOM/CN/90/9044 | Shandong, China | *hsp70* | H1 | JX021428 |
| *L. donovani* | MHOM/CN/86/SC6 | Sichuan, China | *hsp70* | H1 | JX021429 |
| *L. donovani* | IWUI/CN/77/771 | Bachu, China | *hsp70* | H2 | JX021425 |
| *L. donovani* | MHOM/NP/2003/BPK282 | Nepal | *hsp70* | H3 | XM_003862348 |
| *L. donovani* | MHOM/ET/67/HU3 | Ethiopia | *hsp70* | H4 | X52314 |
| *L. donovani* | MHOM/IN/00/DEVI | India | *hsp70* | H3 | FN395028 |
| *L. donovani* | MHOM/MA/95/CRE72 | Morocco | *hsp70* | H3 | HF586352 |
| *L. donovani* | MHOM/SD/87/UGX-MARROW | Sudan | *hsp70* | H3 | HF586386 |
| *L. infantum* | MHOM/EG/87/RTC2 | Egypt | *hsp70* | H1 | HF586350 |
| *L. infantum* | MCAN/IL/97/LRC-L720 | Israel | *hsp70* | H1 | HF586393 |
| *L. infantum* | MHOM/MA/67/ITMAP263 | Morocco | *hsp70* | H1 | FN395033 |
| *L. infantum* | MHOM/PT/00/IMT260 | Portugal | *hsp70* | H1 | FN395032 |
| *L. infantum* | MHOM/MT/85/Buck | Malta | *hsp70* | H1 | FN395031 |
| *L. infantum* | MHOM/MA/67/ITMAP263 | Morocco | *hsp70* | H1 | FN395033 |
| *L. infantum* | MHOM/BR/07/ARL | Brazil | *hsp70* | H1 | FN395037 |
| *L. infantum* | MCAN/ES/98/LLM-877 | Spain | *hsp70* | H1 | XM_001470287 |
| *L. donovani** | MHOM/CN/80/801 | Kashi, China | *hsp70* | H1 | JX970993 |
| *L. donovani** | MHOM/CN/86/SC9 | Sichuan, China | *hsp70* | H1 | JX021430 |
| *L. donovani** | MCAN/CN/97/WDD23 | Gansu, China | *hsp70* | H1 | JX970994 |
| *L. donovani** | MHOM/CN/96/KS6 | Kashgar, China | *hsp70* | H1 | JX970996 |
| *L. braziliensis* | MHOM/CO/90/LEM2216 | Colombia | *hsp70* | H5 | FN395043 |
| *L. guyanensis* | MHOM/PE/02/LH2372 | Peru | *hsp70* | H6 | FN395051 |
| *L. aethiopica* | MHOM/ET/72/L100 | Ethiopia | *hsp70* | H7 | FN395021 |
| *L. tropica* | MHOM/IN/79/DD7 | India | *hsp70* | H8 | FN395025 |
| *L. major* | MHOM/IL/67/LRC-L137 | Israel | *hsp70* | H9 | FN395023 |
| *L. amazonensis* | MHOM/BR/73/M2269 | Brazil | *hsp70* | H10 | EU599090 |
| *L. mexicana* | MNYC/BZ/62/M379 | Belize | *hsp70* | H11 | EU599091 |
| *L. infantum* | MHOM/KE/84/NLB_323 | Kenya | *nagt* | - | DQ836148 |

*(Continued)*

**Table 2.** (Continued)

| Species | WHO code | Origin | Gene | Haplotype number | Accession |
|---------|----------|--------|------|------------------|-----------|
| *L. infantum* | unknown | Turkey | *nagt* | - | AF205934 |
| *L. infantum* | MHOM/CN/50/Bman | China | *nagt* | - | DQ836147 |
| *L. infantum* | MHOM/IR/11/Kazeroun | Iran | *nagt* | - | KF701211 |
| *L. infantum* | MHOM/IR/11/Lamerd1 | Iran | *nagt* | - | KF701212 |
| *L. donovani* | HOM/IN/97/JD | India | *nagt* | - | DQ836150 |
| *L. major* | MHOM/IR/11/Farashband | Iran | *nagt* | - | KF701209 |
| *L. tropica* | MHOM/IR/11/Ghir-Karzin3 | Iran | *nagt* | - | KF701206 |
| *L. mexicana* | HOM/CO/94/1182 | Colombia | *nagt* | - | DQ836161 |
| *L. braziliensis* | HOM/BR/75/M2903 | Brazil | *nagt* | - | DQ836162 |
| *L. turanica* | MRHO/IR/11/Gol-6 | Iran | *nagt* | - | JX103553 |
| *L. gerbilli* | MRHO/IR/10/Gol-9 | Iran | *nagt* | - | JX103531 |

*, designation of the strain does not represent the genotype; -, not available.

## Results

### Sample collection

Samples were collected from 2004 to 2014. In total, 20 *Leishmania* isolates were obtained from patients, sand fly vector (*P. wui*) and Tarim hares (**Table 1**). The collection was carried out during the annual surveys for VL by the Xinjiang Center for Disease Control and Prevention in the AVL-endemic Shufu county in the alluvial plain and in the DT-ZVL-endemic regions in the Bachu, Jiashi and Minfeng counties in the desert area (**Fig 1**). A total of 12 samples were collected from the patients, of which six were under 1 year old from the DT-ZVL-endemic Bachu or Jiashi counties and six were 2 years old or older from the AVL-endemic Shufu county. The demographic data are consistent with the designation of the VL as the two different types indicated. The remaining eight samples were isolated from the vector *P. wui* and Tarim hares in the DT-ZVL-endemic Bachu, Jiashi and Minfeng counties.

### PCR amplification of *nagt*, ITS1 and *hsp70* sequences and phylogenetic analyses

DNA isolated from cultured promastigotes of all 20 samples were subjected to PCR amplification. PCR amplification of the DNA samples with the primer set for *nagt* yielded a single product of the expected size (~1.4 kb) from 13 of the 20 isolates. The reason for the lack of amplification might be caused by several reasons. For instance, *nagt* is a single copy gene which may not have been successfully amplified due to poor sample quality in some cases. These *nagt* sequences were subjected to phylogenetic analysis together with those from representative sequences of other strains/species (Tables 1 and 2). The results clearly showed the segregation of all samples analyzed here into two groups according to the two VL types in the same clade with the reference sequences of the *L. donovani/L. infantum* complex. They were all distant from the other species complexes of the subgenus *Leishmania* (e. g. *L. major*, *L. tropica*, *L. mexicana*) and the subgenus *Viannia* (*L. braziliensis*) (**Fig 2**).

PCR amplification of the same DNAs for the ITS1 locus yielded ~320 bp products from all 20 samples. Sequence analyses of these products separated the 20 samples again into Type 1 and Type 2, in accordance with the two VL types (**Table 1**).

Alignment of Types 1 and 2 ITS1 sequences revealed C/T-substitution at position 78 and A-deletion at location 119 in the latter (**Fig 3**). Type 1 includes all isolates from the 6 patients

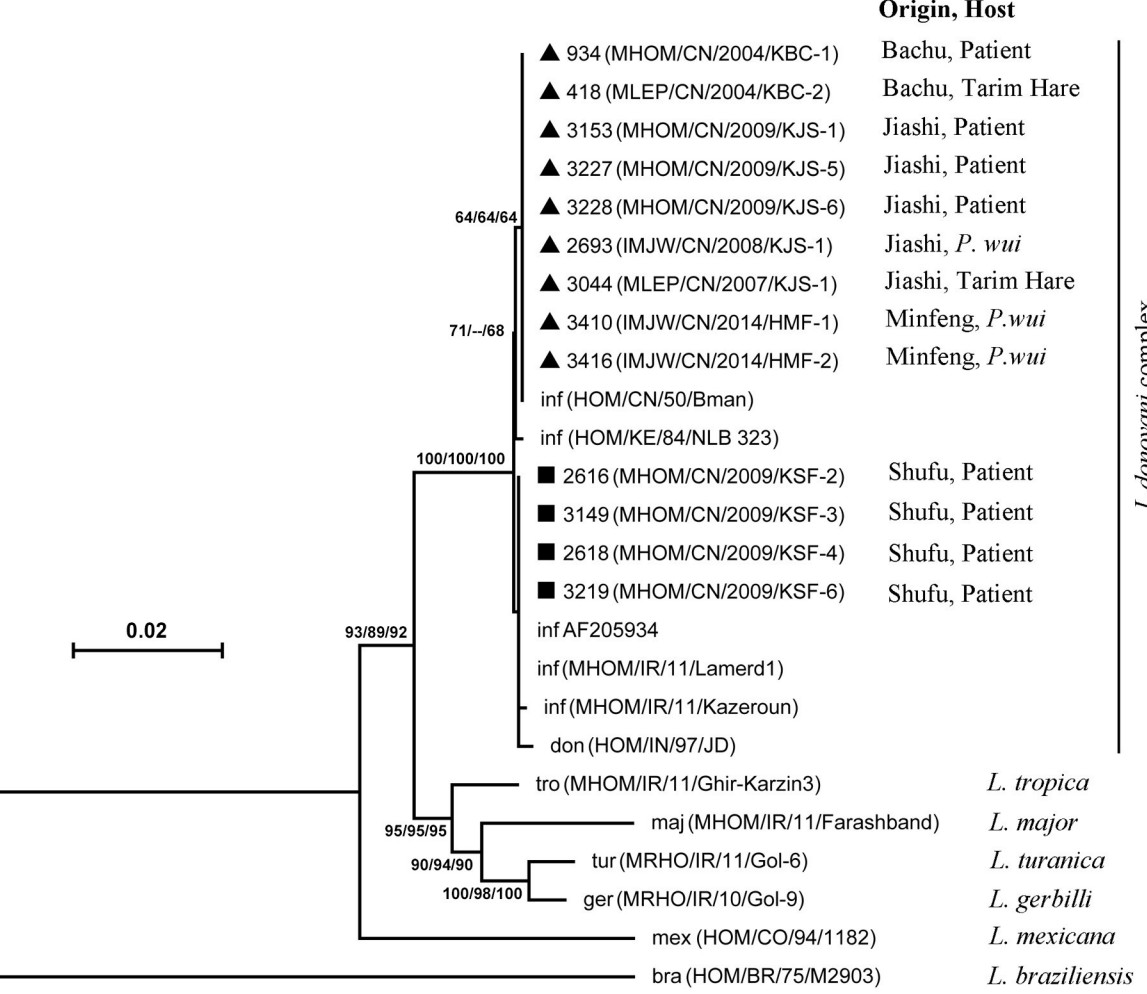

**Fig 2. Phylogenetic tree based on the alignment of amplified section of the *nagt* sequences.** The phylogenetic tree was constructed using the Neighbor-Joining (NJ), Maximum likelihood (ML) and Minimum Evolution (ME) methods, bootstrap values were provided next to nodes (NJ/ML/ME), a value lower than 50 was indicated as (--). Information on the origins and hosts of isolates are provided. The terrain where isolates were collected in this study is indicated with triangle (▲, DT-ZVL foci) or square (■, AVL foci). don, *L. donovani*; inf, *L. infantum*; maj, *L. major*; aet, *L. aethiopia*; tro, *L. tropica*; tur, *L. turanica*; ger, *L. gerbilli*; mex, *L. mexicana*; bra, *L. braziliensis*; *, the species is suggested to be *L. donovani*.

in the AVL-endemic Shufu county, while Type 2 includes the remaining 14 isolates, from patients, the vector *P. wui* and Tarim hares in the DT-ZVL-endemic Bachu, Jiashi and Min-feng counties.

The types 1 and 2 ITS1 sequences were subjected to phylogenetic analyses together with those of the reference strains in the *L. donovani/infantum* complex from other geographical origins (**Table 2**). *L. major* sequence was used as the outgroup (**Fig 4**). All 20 sequences in question were found to group with members of the *L. donovani/infantum* complex in a primary clade, which was then separated into three subclades. One subclade contains all six samples from the AVL foci (Shufu county) that were identical in sequence with those of the *L. infantum* strains that are widely distributed in Mediterranean regions (France, Spain, and Italy), Middle East (Iran), Central Asia (Uzbekistan), and China (Beijing, Xinjiang). Another subclade contains the 14 samples from the DT-ZVL foci that were identical in sequence to those of some Chinese isolates, which were previously typed either as *L. infantum* or

**Fig 3. Multiple sequence alignment of the ITS1 sequences from different haplotypes of Xinjiang *Leishmania* isolates.** The detailed information for each haplotype (I1 to I6) are provided in Tables 1 and 2. The first nucleotide of the ITS1 region that follows the end of the 18s rRNA (NC_007268.2 of the *L. major* strain Friedlin complete genome) was designated as the starting site. Haplotype I6 was chosen as a reference (GenbankAccession: AM901452) and the aligned fragments between position 3 and 121 are shown. Areas shaded in grey show identical sequences while the variable sites were marked with the location of the nucleotide shown above. I1, indicates isolates of AVL foci and *L. infantum* strains. I2, indicates isolates from DT-ZVL foci and *L. donovani*.

*L. donovani*. The third subclade, more distant from the first two, contains samples of mixed origins, ranging from Africa to the Middle East to Asia (China and India).

We successfully PCR- amplified the *hsp70* (~1,286 bp) from 16 of the 20 samples. Alignment of their sequences also revealed differences according to their origin from AVL and DT-ZVL (see **Table 1 for sequence types**). Isolates from the AVL foci had the identical *hsp70* sequence haplotype H1, while isolates from the DT-ZVL foci were all H2. A set of *hsp70* sequences from reference strains, representing both Old and New world species, was retrieved from GenBank (**Table 2**). Phylogenetic analyses of the *hsp70* sequences also support the conclusion that all 20 isolates belong to the *L. donovani/infantum* complex (**Fig 5**). Consistent with the ITS1 analysis, isolates from the AVL foci clustered with *L. infantum* reference strains. In contrast, isolates from the DT-ZVL foci were separated from the isolates of the AVL foci.

Collectively, *Leishmania* isolates from AVL and DT-ZVL in Xinjiang clearly separate into two genetically distinct groups, as determined by sequence analysis of the three different genetic markers. The potential of the Tarim hare as a reservoir for DT-ZVL is strongly suggested by the sequence identity of its isolates with those from patients and vectors for all three phylogenetic markers examined.

## Discussion

In the past, most of the work on leishmaniasis in China has been focused on prevention and control programs. Meanwhile the biological characteristics of the persisting *Leishmania* spp. remains unclear, especially in some of the northwestern regions where VL is still endemic. In this study, two major findings were made from analyses of more recent isolates. Firstly, we demonstrated a clear separation of the AVL and DT-ZVL isolates into 2 different groups in the same *L. donovani/infantum* clade. Secondly, the sequence identity of patient-, vector- and Tarim hare-derived isolates strongly suggests they have a zoonotic transmission cycle and that the Tarim hare acts as a potential reservoir of DT-ZVL. The role of Tarim hares as a reservoir is further supported by the fact that other lagomorphs have been reported in this role for

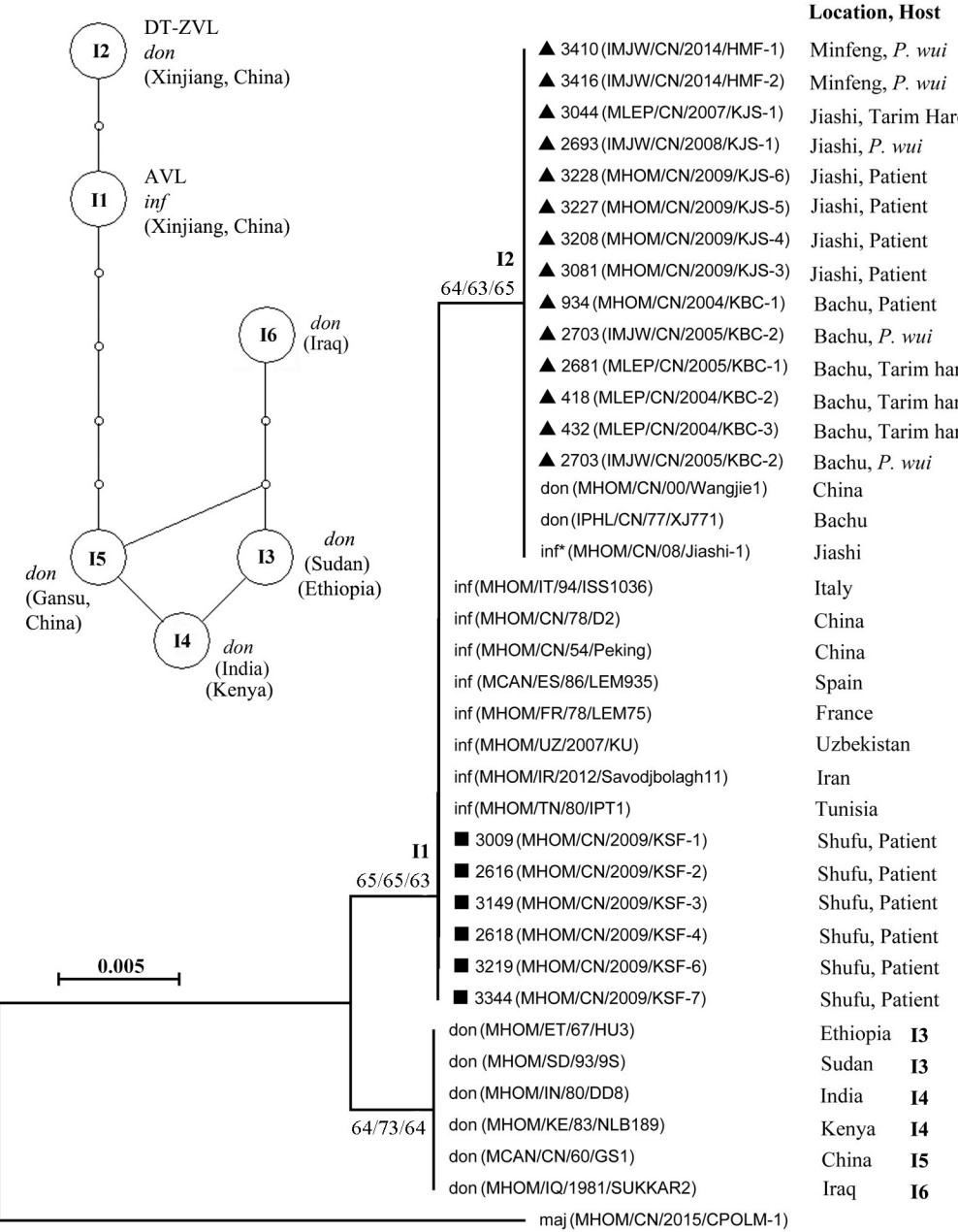

**Fig 4. Statistical parsimony haplotype network and phylogenetic tree based on the ITS1 sequence of the *L. donovani* complex.** The network is generated with TCS software (version 1.21). The haplotypes, I1 to I6, as given in Fig 2, Tables 1 and 2, are represented by an oval circle. Small hollow circles represent unsampled haplotypes and each line represents one mutational step. The phylogenetic tree was constructed using the Neighbor-Joining (NJ), Maximum likelihood (ML) and Minimum Evolution (ME) methods, bootstrap values were provided next to nodes (NJ/ML/ME). The origins and hosts of the isolates are provided. Terrains where isolates were collected in this study are indicated with a triangle (▲, DT-ZVL foci) or a square (■, AVL foci). don, *L. donovani*; inf, *L. infantum*; maj, *L. major*; *, the species is suggested to be *L. donovani*.

zoonotic VL in Spain [24–26]. This also raises the question as to whether lagomorphs should be studied, at the global level, as a potential reservoir for *Leishmania* spp. The pathogen is thought to be transmitted among the natural hosts by sand flies. Humans are occasionally infected when they enter the region where sylvatic infected sand flies exist.

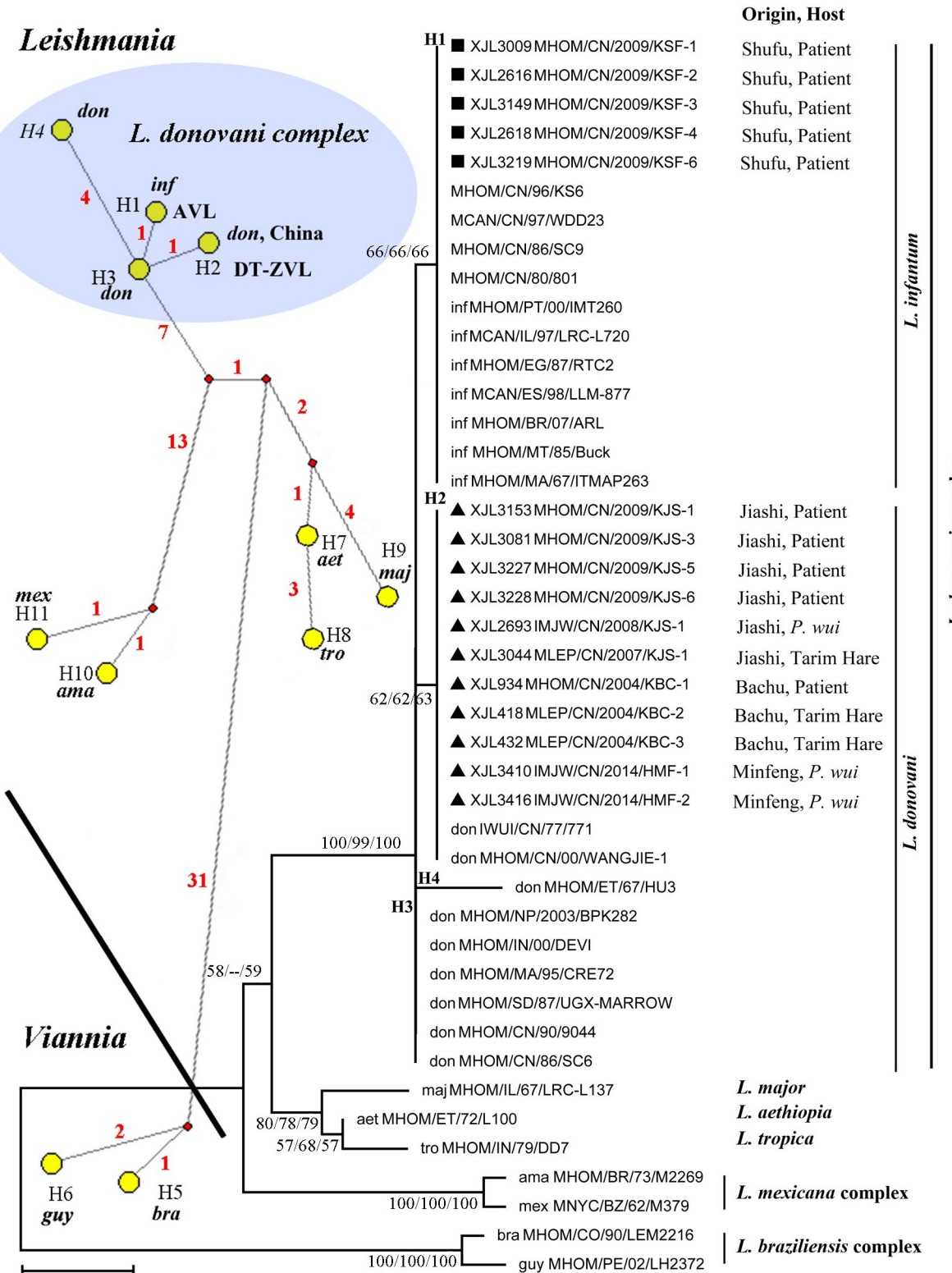

**Fig 5. Haplotype network and phylogenetic tree based on the *hsp70* sequences of the *L. donovani* complex.** A haplotype network constructed using Median Joining and post processing with an MP calculation. The haplotypes of H1 to H11, as given in Fig 2, Tables 1 and 2, are represented by yellow circles. Small red solid circles represent median joining points, and mutational sites between two circles are shown next to the connecting line. The subgenera *Leishmania* and *Viannia* are separated by a thick black line. The phylogenetic tree was

constructed using the Neighbor-Joining (NJ), Maximum likelihood (ML) and Minimum Evolution (ME) methods, bootstrap values were provided next to nodes (NJ/ML/ME), values lower than 50 were indicated as (- -). The origins and hosts of isolates are provided. The terrains where isolates were collected in this study are indicated with triangle (▲, DT-ZVL foci) or square (■, AVL foci). don, *L. donovani*; inf, *L. infantum*; maj, *L. major*; aet, *L. aethiopia*; tro, *L. tropica*; ama, *L. amazonensis*; mex, *L. mexicana*; bra, *L. braziliensis*; guy, *L. guyanensis*; *, the species is suggested to be *L. donovani*. (A total of 1047 NT between sites 590–1636 were analyzed in the final data set and are numbered from the start codon of the *hsp70* gene). Note: MHOM/CN/00/Wangjie1 and IWUI/CN/77/771 are heterozygous and only H2 was analyzed. H1: *Leishmania infantum* and isolates from AVL; H2: isolates from DT-ZVL. Other details of each haplotype are provided in Tables 1 and 2.

Our study has provided preliminary evidence for the genetic difference in *Leishmania* isolates from the AVL and DT-ZVL regions in Xinjiang. In our study, we used genetic markers which were either single copy (*nagt*) or multicopy (ITS1 and *hsp70*). The latter proved more successful for amplification due to their multicopy nature when used for these challenging samples. However, discriminatory markers or typing methods with a higher resolution such as multilocus microsatellite typing (MLMT) or multilocus sequence typing (MLST) are encouraged to be considered for further analyses [6,27]. Ideally, whole genome sequencing would provide the ultimate solution to identify all possible genetic factors correlated with adaptation to different disease conditions [28–30]. Another possible limitation in this study was that only cultivable parasites were used for the analyses. These might represent a bias in the parasite population studied. Thus, phylogenomic analysis of lesion-derived amastigotes from mammalian hosts or promastigotes from sand-fly gut material will be a necessity for further study. When we compared published *nagt*, ITS1 and *hsp70* sequences (MHOM/IN/1983/AG83; MHOM/IN/00/DEVI; MHOM/IN/80/DD8) from Indian *L. donovani* clinical samples, they were 100% identical to the Indian representative sequences we used in this study–we did not, therefore, include these in our phylogenetic study. To confirm the role of the Tarim hare as an animal reservoir, further detailed studies are required which need to involve collection of a larger sample set of infected animals (and human hosts) and analyse lesion-derived amastigotes with the multilocus markers or genomics analyses described above. However, due to the comprehensive control of leishmaniasis in China, both patient and animal infections are rare, and therefore some of the questions posed above may not be able to be answered.

There is little doubt that *Leishmania* spp. causing persistent VL in Xinjiang belongs to the *L. donovani* complex, but we found an atypical association between *L. infantum*/*donovani* and the epidemiology of the ZVL/AVL types. For the isolates from Shufu county, a well-known foci of AVL [3], sequence analysis indicated that they are, most likely, *L. infantum*. This is unusual, considering that the disease type is AVL. First, these isolates were mainly found in patients over 2 y.o. (Table 1), which is considered the main characteristic of *L. donovani* infection [31]. In addition, no animal reservoir has been found for the Xinjiang *L. infantum* species. A possibility that a hidden ZVL co-exists with AVL in the Shufu county was discounted, since all 6 *Leishmania* isolates in this region display the same ITS1 and *hsp70* haplotype. The existence of *L. infantum* in AVL has also been reported in other studies. A previously investigated isolate from Kashi City (an AVL focus), MHOM/CN/80/801, with an identical *hsp70* sequence to the *L. infantum* clade, was suggested to be *L. infantum* by MLST in another study [6]. In addition, several adult cases of *L. infantum* infection were reported in Spain where *L. infantum* was not considered to be a local species [32]. Thus, we conclude that the Xinjiang AVL isolates from the oases of the Kashgar county are *L. infantum* with atypical clinical manifestations and no identified animal reservoir.

On the other hand, all 14 isolates from the DT-ZVL foci, i.e. Bachu, Jiashi and Minfeng counties, are genetically close to *L. donovani*, even though they are responsible for zoonotic disease. The ITS1 phylogenetic tree showed a similar tree topology to a previous study [8], in

which all the isolates formed a sub-clade within the *L. infantum* cluster. However, further analysis revealed that their ITS1 sequences were identical to the MLEE-typed strain *L. donovani* MHOM/CN/00/Wangjie1, and also the strain IPHL/CN/77/XJ771 from Bachu county, which were deemed previously to have identity with *L. donovani* [9,13]. Additionally, our *hsp70* data also suggested that these isolates are Xinjiang-specific *L. donovani* strains that are genetically close to the common *L. donovani* strains (H3) from India, Sudan, Nepal and Morocco in the haplotype network (Fig 5).

*L. donovani* is considered to be mainly anthroponotic while *L. infantum* is zoonotic with dogs serving as a primary reservoir. There have been no reports of AVL caused by *L. infantum*, while ZVL caused by *L. donovani* has been documented before. Thus, all the Xinjiang *Leishmania* isolates we have studied display atypical epidemiological features. This suggests that more attention needs to be paid when classifying these species on clinical grounds, since there might also be underreporting or mis-reporting occurrences elsewhere.

In conclusion, species of the *L. donovani* complex are responsible for AVL and DT-ZVL in Xinjiang Autonomous Region of China. We consider that the two types of VL are caused by two different groups of parasite. Epidemiological conditions have a great impact on shaping the endemic area occupied by these parasites. Our results further support that the Tarim hare is most likely the reservoir for *L. donovani* and the source of infection in the desert region. Further control measures targeting these wild animals may be needed for the effective control of this disease. More discriminatory methods, particularly direct whole genome sequencing of parasites from host tissues, will be the preferred approach to clearly dissect the complicated situation in these Chinese leishmania parasites.

## Acknowledgments

We would like to deeply thank Professor K.-P. Chang of Chicago Medical School/Rosalind Franklin University of Medicine and Science, North Chicago, USA for his kind encouragements to carry out this work.

## Author Contributions

**Conceptualization:** Li-Fu Liao, De-Hua Lai, Zhao-Rong Lun.

**Formal analysis:** Yun-Fu Chen, Jiang-Mei Gao, Yan-Zi Wen, De-Hua Lai.

**Funding acquisition:** Yan-Zi Wen, Zhao-Rong Lun.

**Investigation:** Yun-Fu Chen, Li-Fu Liao, Na Wu, Jiang-Mei Gao.

**Methodology:** De-Hua Lai, Zhao-Rong Lun.

**Supervision:** De-Hua Lai, Zhao-Rong Lun.

**Validation:** Peng Zhang, De-Hua Lai.

**Visualization:** Yun-Fu Chen, Peng Zhang, Yan-Zi Wen, De-Hua Lai, Zhao-Rong Lun.

**Writing – original draft:** Yun-Fu Chen, De-Hua Lai, Zhao-Rong Lun.

**Writing – review & editing:** Li-Fu Liao, Peng Zhang, Yan-Zi Wen, Geoff Hide, De-Hua Lai, Zhao-Rong Lun.

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
