## [Decision Letter · Decision Letter 0]

29 Aug 2021

Dear Mr Lai,

Thank you very much for submitting your manuscript "Species identification and phylogenetic analysis of Leishmania isolated from patients, vectors and hares in the Xinjiang Autonomous Region, The People's Republic of China" for consideration at PLOS Neglected Tropical Diseases. As with all papers reviewed by the journal, your manuscript was reviewed by members of the editorial board and by several independent reviewers. In light of the reviews (below this email), we would like to invite the resubmission of a significantly-revised version that takes into account the reviewers' comments. 

All reviewers came to the conclusion that this is an very interessant study, which has however some restrictions e.g. small sample size and limited number of loci studied. I think the authors should try to address these comments in the revised manuscript. For instance comparison of their data to data from Indian L. donovani as suggested by reviewer 2 seems to be feasible. I also agree with reviewer 1 that it would be very interesting to perform whole-genome sequencing of these Chinese strains and to compare these data with the numerous L. donovani/ L. infantum sequences already available . But this could be perhaps w be published in a next manuscript. The present study could be considered as a pilot study. The authors need however to discuss the restrictions and future prospects of their work in this submission.

We cannot make any decision about publication until we have seen the revised manuscript and your response to the reviewers' comments. Your revised manuscript is also likely to be sent to reviewers for further evaluation.

Sincerely,

Gabriele Schönian

Associate Editor

Michael Boshart

Deputy Editor

All reviewers came to the conclusion that this is an very interessant study, which has however some restrictions e.g. small sample size and limited number of loci studied. I think the authors should try to address these comments in the revised manuscript. For instance comparison of their data to data from Indian L. donovani as suggested by reviewer 2 seems to be feasible. I also agree with reviewer 1 that it would be very interesting to perform whole-genome sequencing of these Chinese strains and to compare these data with the numerous L. donovani/ L. infantum sequences already available . But this could be perhaps w be published in a next manuscript. The present study could be considered as a pilot study. The authors need however to discuss the restrictions and future prospects of their work in this submission.

Reviewer's Responses to Questions

**Key Review Criteria Required for Acceptance?**

**Methods**

-Are the objectives of the study clearly articulated with a clear testable hypothesis stated?

-Is the study design appropriate to address the stated objectives?

-Is the population clearly described and appropriate for the hypothesis being tested?

-Is the sample size sufficient to ensure adequate power to address the hypothesis being tested?

-Were correct statistical analysis used to support conclusions?

-Are there concerns about ethical or regulatory requirements being met?

Reviewer #1: With cultures available, it may be relatively easy to sequence genomes of all these 20 isolates. The downstream analysis would make all the conclusions stronger. The presented analyses (based on 3 loci, some of which were not successfully amplified across the dataset - f.e. nagt) should be considered preliminary.

Reviewer #2: Objective is clear as stated

The sample size is a bit small

Sequence phylogenetic analysis based on MEGA version 5

Met ethical/regulatory requirements

Reviewer #3: The manuscript “Species identification and phylogenetic analysis of Leishmania isolated from patients, vectors and hares in the Xinjiang Autonomous Region, The People's Republic of China” applies a molecular epidemiological approach on 20 new field isolates obtained from two endemic sites, which are analysed for the three diagnostic genes ITS1, HSP70 and NAGT by PCR amplification, sequencing and phylogenetic mapping. The manuscript provides important new epidemiological information by (i) distinguishing both L. infantum and L. donovani infection in the Xinjiang Autonomous Region, (ii) demonstrating a non-classical, anthroponotic transmission cycle for L. infantum and a non-classical, zoonotic transmission cycle for L. donovani isolates, and (iii) involving hares as novel reservoir for L. donovani. Overall, the paper is very well written, and the drawn conclusions are fully supported by the results. Thus, I do not have any major criticism. One minor comment is that often the authors write that leishmaniasis is transmitted by sand flies, yet it is the parasite and not the disease that is transmitted. Also, the authors may consider combining the haplotype networks with the corresponding phylogenetic analyses rather than showing the first ones in separate figures.

**Results**

-Does the analysis presented match the analysis plan?

-Are the results clearly and completely presented?

-Are the figures (Tables, Images) of sufficient quality for clarity?

Reviewer #1: The presented results should be further complemented (or replaced altogether) by the analyses based on whole genome sequences.

Reviewer #2: Phylogenetic analyses are standard

Figures and Tables are acceptable

Reviewer #3: The manuscript “Species identification and phylogenetic analysis of Leishmania isolated from patients, vectors and hares in the Xinjiang Autonomous Region, The People's Republic of China” applies a molecular epidemiological approach on 20 new field isolates obtained from two endemic sites, which are analysed for the three diagnostic genes ITS1, HSP70 and NAGT by PCR amplification, sequencing and phylogenetic mapping. The manuscript provides important new epidemiological information by (i) distinguishing both L. infantum and L. donovani infection in the Xinjiang Autonomous Region, (ii) demonstrating a non-classical, anthroponotic transmission cycle for L. infantum and a non-classical, zoonotic transmission cycle for L. donovani isolates, and (iii) involving hares as novel reservoir for L. donovani. Overall, the paper is very well written, and the drawn conclusions are fully supported by the results. Thus, I do not have any major criticism. One minor comment is that often the authors write that leishmaniasis is transmitted by sand flies, yet it is the parasite and not the disease that is transmitted. Also, the authors may consider combining the haplotype networks with the corresponding phylogenetic analyses rather than showing the first ones in separate figures.

**Conclusions**

-Are the conclusions supported by the data presented?

-Are the limitations of analysis clearly described?

-Do the authors discuss how these data can be helpful to advance our understanding of the topic under study?

-Is public health relevance addressed?

Reviewer #1: See above. I believe the phylogenomic analyses will make the manuscript much stronger.

Reviewer #2: Conclusions acceptable, but weak due to small sample size

Discussion includes statements for the limitations of the analyses

Significance of the data obtained was discussed

Public health is not part of the work, but relevant points mentioned

Reviewer #3: The manuscript “Species identification and phylogenetic analysis of Leishmania isolated from patients, vectors and hares in the Xinjiang Autonomous Region, The People's Republic of China” applies a molecular epidemiological approach on 20 new field isolates obtained from two endemic sites, which are analysed for the three diagnostic genes ITS1, HSP70 and NAGT by PCR amplification, sequencing and phylogenetic mapping. The manuscript provides important new epidemiological information by (i) distinguishing both L. infantum and L. donovani infection in the Xinjiang Autonomous Region, (ii) demonstrating a non-classical, anthroponotic transmission cycle for L. infantum and a non-classical, zoonotic transmission cycle for L. donovani isolates, and (iii) involving hares as novel reservoir for L. donovani. Overall, the paper is very well written, and the drawn conclusions are fully supported by the results. Thus, I do not have any major criticism. One minor comment is that often the authors write that leishmaniasis is transmitted by sand flies, yet it is the parasite and not the disease that is transmitted. Also, the authors may consider combining the haplotype networks with the corresponding phylogenetic analyses rather than showing the first ones in separate figures.

**Editorial and Data Presentation Modifications?**

Reviewer #1: N/A

Reviewer #2: P. 3, last para., 1st sentence: Clarify the description of DT-ZVL, which has been categorized into sandy desert type and pebble desert type with different sand fly vectors in Xinjiang.

P. 4, lines 5-6: Define what are "Leishmania antigens" and "typical VL symptoms" besides splenomegaly ?

P. 5, last 3 lines: "unknown reasons" and poor "sample quality" are inconsistent statements

P. 6, 3rd para., line 1: "A/G substitution" is inconsistent with nts shown in position 78 in Fig. 3 ? 

P. 6, last para., 1st line: Is hsp70 encoded by tandem-repeated genes in Leishmania ? If so, the negative PCR is surprising for 4 of the 20 samples examined. 

P. 8, 2nd para., line 5-8: Several points here need attention for clarification. L. infantum has long been considered as the causative species of AVL in the Kashi alluvial plain based on DNA/sequence analyses. It is incorrect to say "no animal reservoir has been found for the Chinese L. infantum". Dog has been clearly shown as a reservoir and racoon dog considered as another one for infantum VL in eastern China. It may be acceptable to replace "Chinese" with "Xinjiang" in that statement. 

P. 8, last sentence: The same problem as stated above. Negative finding in search for animal reservoirs also cannot be conclusive. 

P. 9, 1st para. line 10 and 2nd para., line 4: Replace "China" and "Chinese" with Xinjiang in these two statement to avoid the same problem.

P. 10,Line 8: "bug" "Largus largus" ? Do you mean Lagurus lagurus ? If so, it is not a bug !?

Reviewer #3: The manuscript “Species identification and phylogenetic analysis of Leishmania isolated from patients, vectors and hares in the Xinjiang Autonomous Region, The People's Republic of China” applies a molecular epidemiological approach on 20 new field isolates obtained from two endemic sites, which are analysed for the three diagnostic genes ITS1, HSP70 and NAGT by PCR amplification, sequencing and phylogenetic mapping. The manuscript provides important new epidemiological information by (i) distinguishing both L. infantum and L. donovani infection in the Xinjiang Autonomous Region, (ii) demonstrating a non-classical, anthroponotic transmission cycle for L. infantum and a non-classical, zoonotic transmission cycle for L. donovani isolates, and (iii) involving hares as novel reservoir for L. donovani. Overall, the paper is very well written, and the drawn conclusions are fully supported by the results. Thus, I do not have any major criticism. One minor comment is that often the authors write that leishmaniasis is transmitted by sand flies, yet it is the parasite and not the disease that is transmitted. Also, the authors may consider combining the haplotype networks with the corresponding phylogenetic analyses rather than showing the first ones in separate figures.

**Summary and General Comments**

Reviewer #1: In my opinion, this is a very interesting manuscript and this is a good fit for the PLoS Neglected Tropical Diseases Journal. I would strongly recommend to enforce it by whole-genome sequencing of the isolates under study and subsequent phylogenomic analysis. From the point of view of a classical parasitologist, the story is extremely important.

Reviewer #2: The work presented is highly significant by studying Leishmania rarely examined in an endemic area of VL with unique epidemiology. The authors correctly pointed out in the Discussion that amastigotes should be examined for genomic phylogenetic analyses. Such data from amastigotes of Indian L donovani clinical samples have been published. Their hsp70, ITS1 and nagt sequences should be included for phylogenetic analysis in this study. The relative merits of the three sequence markers used should be discussed with reference to their copy number per haploid genome. It is further suggested to include a brief discussion for future experimental study to verify the role of Tarim hare as a reservoir of DT-ZVL.

Reviewer #3: The manuscript “Species identification and phylogenetic analysis of Leishmania isolated from patients, vectors and hares in the Xinjiang Autonomous Region, The People's Republic of China” applies a molecular epidemiological approach on 20 new field isolates obtained from two endemic sites, which are analysed for the three diagnostic genes ITS1, HSP70 and NAGT by PCR amplification, sequencing and phylogenetic mapping. The manuscript provides important new epidemiological information by (i) distinguishing both L. infantum and L. donovani infection in the Xinjiang Autonomous Region, (ii) demonstrating a non-classical, anthroponotic transmission cycle for L. infantum and a non-classical, zoonotic transmission cycle for L. donovani isolates, and (iii) involving hares as novel reservoir for L. donovani. Overall, the paper is very well written, and the drawn conclusions are fully supported by the results. Thus, I do not have any major criticism. One minor comment is that often the authors write that leishmaniasis is transmitted by sand flies, yet it is the parasite and not the disease that is transmitted. Also, the authors may consider combining the haplotype networks with the corresponding phylogenetic analyses rather than showing the first ones in separate figures.

PLOS authors have the option to publish the peer review history of their article (what does this mean?). If published, this will include your full peer review and any attached files.

Reviewer #1: No

Reviewer #2: No

Reviewer #3: No
---

## [Decision Letter · Decision Letter 1]

4 Dec 2021

Dear Dr Lai,

We are pleased to inform you that your manuscript 'Species identification and phylogenetic analysis of Leishmania isolated from patients, vectors and hares in the Xinjiang Autonomous Region, The People's Republic of China' has been provisionally accepted for publication in PLOS Neglected Tropical Diseases.

Best regards,

Gabriele Schönian

Associate Editor

Michael Boshart

Deputy Editor

Ther are no further comments to the authors

Reviewer's Responses to Questions

**Key Review Criteria Required for Acceptance?**

**Methods**

-Are the objectives of the study clearly articulated with a clear testable hypothesis stated?

-Is the study design appropriate to address the stated objectives?

-Is the population clearly described and appropriate for the hypothesis being tested?

-Is the sample size sufficient to ensure adequate power to address the hypothesis being tested?

-Were correct statistical analysis used to support conclusions?

-Are there concerns about ethical or regulatory requirements being met?

Reviewer #1: (No Response)

Reviewer #3: The authors have responded to my queries.

**Results**

-Does the analysis presented match the analysis plan?

-Are the results clearly and completely presented?

-Are the figures (Tables, Images) of sufficient quality for clarity?

Reviewer #1: (No Response)

Reviewer #3: The authors have responded to my queries.

**Conclusions**

-Are the conclusions supported by the data presented?

-Are the limitations of analysis clearly described?

-Do the authors discuss how these data can be helpful to advance our understanding of the topic under study?

-Is public health relevance addressed?

Reviewer #1: (No Response)

Reviewer #3: The authors have responded to my queries.

**Editorial and Data Presentation Modifications?**

Reviewer #1: (No Response)

Reviewer #3: The authors have responded to my queries.

**Summary and General Comments**

Reviewer #1: (No Response)

Reviewer #3: The authors have responded to my queries.

PLOS authors have the option to publish the peer review history of their article (what does this mean?). If published, this will include your full peer review and any attached files.

Reviewer #1: No

Reviewer #3: No

---

## [Editor Report · Acceptance letter]

10 Dec 2021

Dear Dr Lai,

We are delighted to inform you that your manuscript, "Species identification and phylogenetic analysis of Leishmania isolated from patients, vectors and hares in the Xinjiang Autonomous Region, The People's Republic of China," has been formally accepted for publication in PLOS Neglected Tropical Diseases.

Best regards,

Shaden Kamhawi

co-Editor-in-Chief

Paul Brindley

co-Editor-in-Chief
